# Lipoprotein Deprivation Reveals a Cholesterol-Dependent Therapeutic Vulnerability in Diffuse Glioma Metabolism

**DOI:** 10.3390/cancers14163873

**Published:** 2022-08-11

**Authors:** James Wood, Salah Abdelrazig, Sergey Evseev, Catherine Ortori, Marcos Castellanos-Uribe, Sean T. May, David A. Barrett, Mohammed Diksin, Sajib Chakraborty, Dong-Hyun Kim, Richard G. Grundy, Ruman Rahman

**Affiliations:** 1Children’s Brain Tumour Research Centre, School of Medicine, Biodiscovery Institute, University of Nottingham, Nottingham NG7 2RD, UK; 2Centre for Analytical Bioscience, Advanced Materials and Healthcare Technologies Division, School of Pharmacy, University of Nottingham, Nottingham NG7 2RD, UK; 3Nottingham Arabidopsis Stock Centre, School of Biosciences, University of Nottingham, Nottingham NG7 2RD, UK; 4Department of Biochemistry, University of Dhaka, Dhaka 1000, Bangladesh

**Keywords:** diffuse glioma, metabolism, lipoprotein, cholesterol, LXR agonists

## Abstract

**Simple Summary:**

High-grade gliomas are aggressive cancers that arise in children and adults, for which there is an urgent need for more effective drug therapies. Targeting the energy requirements (‘metabolism’) of these cancer cells may offer a new avenue for therapy. Cholesterol is a fatty substance found on the surface of cancer cells. Our research shows that childhood high-grade gliomas require cholesterol for their energy needs. By repurposing a drug called LXR-623 to reduce the levels of cholesterol inside high-grade glioma cancer cells, we could impair the growth of these cells in laboratory conditions. These results provide evidence for future experiments using LXR-623 to test whether this drug is able to increase the survival of mice with similar high-grade gliomas.

**Abstract:**

Poor outcomes associated with diffuse high-grade gliomas occur in both adults and children, despite substantial progress made in the molecular characterisation of the disease. Targeting the metabolic requirements of cancer cells represents an alternative therapeutic strategy to overcome the redundancy associated with cell signalling. Cholesterol is an integral component of cell membranes and is required by cancer cells to maintain growth and may also drive transformation. Here, we show that removal of exogenous cholesterol in the form of lipoproteins from culture medium was detrimental to the growth of two paediatric diffuse glioma cell lines, KNS42 and SF188, in association with S-phase elongation and a transcriptomic program, indicating dysregulated cholesterol homeostasis. Interrogation of metabolic perturbations under lipoprotein-deficient conditions revealed a reduced abundance of taurine-related metabolites and cholesterol ester species. Pharmacological reduction in intracellular cholesterol via decreased uptake and increased export was simulated using the liver X receptor agonist LXR-623, which reduced cellular viability in both adult and paediatric models of diffuse glioma, although the mechanism appeared to be cholesterol-independent in the latter. These results provide proof-of-principle for further assessment of liver X receptor agonists in paediatric diffuse glioma to complement the currently approved therapeutic regimens and expand the options available to clinicians to treat this highly debilitating disease.

## 1. Introduction

Glioblastoma isocitrate dehydrogenase (IDH) wild-type (GBM) and paediatric-type high-grade gliomas are diffuse malignant gliomas recently reclassified by the World Health Organisation (WHO), which present a formidable clinical challenge in adults and children, respectively, given its highly aggressive nature [1]. Despite advances in cancer therapy and management, survival rates remain low, with 2-year survival rates of 23.5% and 34.3% in the 20+ and <19 years age groups, respectively, within the US population [2]. Microarray-based transcriptomic studies into adult and paediatric diffuse gliomas have distinguished different underlying tumour biologies [3], propelling research into genome-directed targeted therapies [4]. To date, the implementation of personalised medicine strategies has yet to obtain phase II clinical trial efficacy and markedly extended survival, beyond that achieved by concurrent and adjuvant chemotherapy with radiotherapy after surgical excision [5]. This failing is attributed to resistance mechanisms inherent to diffuse gliomas, such as intratumour heterogeneity [6], coupled to the infiltrative nature of the disease. Given the redundancy of growth-promoting signalling pathways, impetus to examine the downstream metabolic pathways on which these input signals converge has shown promise as an alternative and/or complementary adjunct to conventional chemotherapy. Recently, in vitro and in vivo studies of EGFRvIII-expressing GBM models revealed a non-oncogene addiction to lipoproteins, due to the activation of the PI3K/AKT–SREBP–LDLR signalling axis, highlighting a metabolic ‘Achilles heel’ of tumour specificity related to the homeostatic control of intracellular cholesterol levels [7,8,9].

Lipid rafts are assemblies of lipids and proteins that manifest on mammalian cell membranes via dense packaging of molecules mediated by cholesterol [10]. High-density lipoprotein (HDL) levels observed clinically [11,12,13] and high cholesterol content within tumour membranes [14] have established a link between tumourigenesis and cholesterol metabolism. Receptor-mediated uptake of circulating low-density lipoproteins (LDLs) via LDL receptors (LDLR) facilitates intracellular LDL levels [15]. Although transfer of LDLs into the central nervous system (CNS) is prevented by the blood–brain barrier, relatively smaller HDL molecules can efficiently penetrate into the brain, whilst other apolipoprotein E (ApoE)-containing HDLs are synthesised by astrocytes. Lipoproteins within the CNS are stipulated to transfer cholesterol and phospholipids between astrocytes and neurons, but also have documented roles in the pathogenesis of Alzheimer’s disease [16].

Early work in GBM identified high cholesterol requirements associated with increased LDLR binding activity, although some tumours showed a preference for de novo synthesis [17]. Combined with the demonstration of lipoprotein dependency in U87MG cells and Ras-transformed astrocytes for sustained proliferation [18], these studies highlight tumour growth-promoting properties of lipoproteins in GBM. Here, we demonstrate lipoprotein dependency as a key feature in paediatric diffuse gliomas and identify cholesterol metabolism as a vulnerability to be therapeutically targeted using liver X receptor (LXR) agonists in both the adult and paediatric setting.

## 2. Materials and Methods

### 2.1. Cell Culture and Cellular Viability Assessment

U373 (adult IDH wild-type GBM) cell line was purchased from American Type Culture Collection (ATCC No. HTB-17). KNS42 (paediatric diffuse hemispheric glioma, histone 3 G34-mutant) and SF188 (paediatric diffuse histone 3 wild-type high-grade glioma), UW479 (paediatric glioma grade III), Res259 (paediatric glioma grade II) and Res186 (paediatric glioma grade I) cell lines were a kind gift from Prof. Chris Jones (Institute of Cancer Research, London, UK). Cell lines were grown in Dulbecco’s Modified Eagle Medium (DMEM) supplemented with 10% foetal bovine serum (FBS), non-essential amino acids (NEAAs) and penicillin/streptomycin (P/S) at 37 °C in a 5% CO_2_ humidified environment. Cells cultured under lipoprotein-deficient conditions were grown in media containing lipoprotein-deficient serum from foetal bovine (LD-FBS; Sigma, Gillingham, UK) in place of FBS. Spheroids were grown in ultra-low attachment 96-well plates (Corning, Flintshire, UK; 7007) to a diameter of approximately 300 µm after four days culture, at which point experimental conditions were implemented (n = 6 per condition). Brightfield images were taken using a Canon camera and spheroid area (and hence diameter) was measured using a macro developed by Ivanov et al. (2014) [19] for use within ImageJ (FiJi version 1.0, created by Wayne Rasband, Github, Bethesda, MD, USA). Therapeutic evaluation of LXR-623 (WAY-252623; Selleckchem, Cambridge, UK) was performed in 2D 96-well format using DMEM supplemented with 1% FBS, non-essential amino acids and penicillin/streptomycin. The drug dissolved in dimethyl sulfoxide (DMSO) (Sigma, Gillingham, UK) was added at dose concentrations ranging up to 30 µM for 72 h. All cell viability assessments were conducted using the resazurin-based PrestoBlue reagent (ThermoFisher Scientific, Horsham, UK; A13261), with fluorescence measured at wavelengths of 544-nm excitation and 590-nm emission using a FLUOstar Omega microplate reader (BMG LABTECH, Aylesbury, UK). For crystal violet staining, cells were grown in 48-well plates at appropriate seeding densities. Growth under lipoprotein-replete or -deplete conditions was assessed by staining cells with a 0.2% crystal violet (Sigma; C6158) solution for 30 min, followed by a series of washes with distilled water to remove excess stain. Crystal violet was solubilised with a 1% sodium dodecyl sulphate (MP Biomedicals, Loughborough, UK) solution, before absorbance was measured at 570 nm using a FLUOstar Omega microplate reader (BMG LABTECH). The procedure was repeated with empty wells as controls to remove any background signal.

### 2.2. Flow Cytometric Cell Cycle Analysis

Cell lines were cultured under lipoprotein-replete/deplete conditions or treated with LXR-623 at the IC_50_ concentration or at a higher dose required to achieve a 75% reduction in cellular viability after 72 h. Cells were then trypsinised, suspended in phosphate buffered saline (PBS) and fixed in ethanol cooled to 4 °C for at least 2 h. Fixed cells were then resuspended in propidium iodide (PI) staining solution (0.1% (*v*/*v*) Triton X-100, 10 µg/mL PI, and 100 µg/mL DNase-free RNase A in PBS) and kept in the dark at room temperature for 30 min. PI binding to DNA was measured using the Beckman Coulter FC500 flow cytometer, with excitation and emission wavelengths set to 536 and 617 nm, respectively, and processed using Weasel software.

### 2.3. Transcriptomics

RNA from the cells cultured under lipoprotein-replete/deplete conditions or treated with the IC_50_ concentrations of LXR-623 for 72 h was extracted and assessed for quality on the Agilent Bioanalyzer 2100 instrument, (Agilent, Stockport, UK) with RNA integrity number (RIN) scores of >7 as the threshold for downstream microarray analysis using Affymetrix Human Gene ST2.1 Strips. Evaluation of array hybridisation was conducted using the Partek Genomics Suite to identify chip artefacts. Differential gene expression analysis was conducted in R using *limma* [20], and network analysis was performed using NetworkAnalyst [21].

### 2.4. Survival Analyses

The hazard ratios for the top differentially expressed genes (*STC2*, *CA9*, *BNIP3*, *VEGFRA*, and *NDNF*) in U373 cells upon lipoprotein deprivation were estimated by Cox proportional hazards model regression analysis, based on The Cancer Genome Atlas (TCGA) adult GBM cohort (https://tcga-data.nci.nih.gov/tcga/). Analysis was performed at a 95% confidence interval. To investigate the impact of gene-signatures on survival outcomes, a multivariate Cox analysis based on TCGA mRNA Log_2_ expression data (TPM + 1) of gene signatures was performed using GEPIA2. Briefly, the survival curves were generated using Kaplan–Meier analysis for overall survival by using a median survival cutoff. Survival analysis for *STC2*, *CA9*, *BNIP3*, *VEGFRA*, and *NDNF* was performed separately. For classifying the high- and low-expression cohorts, the median expression threshold was set (cut-off high: upper 50% and cut-off low: lower 50%). 

### 2.5. Metabolomics Data Acquisition and Analysis

Bi-phasic extraction of metabolites was conducted using a 1:3:1 ratio of methanol:chloroform:water. Polar metabolites were separated using a zwitterionic-polymeric hydrophilic interaction liquid chromatography (ZIC-pHILIC) column (150 × 4.6 mm, 5 µm particle size), maintained at 45 °C in a ThermoFisher Accela LC system (Thermo Fisher Scientific, Horsham, UK). A linear LC gradient from 80% B to 5% B was used over 15 min, followed by a 5 min wash with 5% B and 7 min re-equilibration with 80% B at a flow rate of 300 µL/minute, where B was 100% acetonitrile and A was 20 mM ammonium carbonate in 18.2 MΩ water (Elga Maxima; Elga LabWater, High Wycombe, UK). Injection volume was set to 10 µL and samples were maintained at 4 °C. Mass spectrometry (MS) was performed using an Exactive Orbitrap MS (Thermo Fisher Scientific, UK), with a HESI-II probe operated in a polarity switching mode. The MS spray voltage was 4.5 kV (ESI+), 3.5 (ESI−), capillary voltage 20 V (ESI+), −15 V (ESI−), sheath, auxiliary and sweep gas flow rates were 40, 5 and 1 (arbitrary unit), respectively, for both modes. Heater and capillary temperature were maintained at 150 °C and 275 °C, respectively. Data were acquired using Xcalibur software (Thermo Fisher Scientific, Hemel Hampstead, UK) in full scan mode, with a resolution of 50,000 from *m*/*z* 70–1400 at 4 Hz scan rate. Raw liquid chromatography (LC)–MS data were processed with XCMS for untargeted peak-picking [22] and mzMatch for alignment and annotation of the related peaks [23]. IDEOM software (version 1.0, University of Glasgow, Glasgow, UK) was used for noise filtering and putative metabolite identification, as shown previously [24,25]. Metabolite identification was performed by matching accurate masses and retention times of authentic standards (Level 1 metabolite identification according to the metabolomics standards initiative [26,27], but when standards were not available, predicted retention times were employed; hence, these identifications should be considered as putative (Level 2 identification)).

### 2.6. Lipidomics Data Acquisition and Analysis

Lipidomics analysis of the non-polar extraction phase used a reversed phase ACE Excel 2 C18 column (50 × 2.1 mm, 2 µm particle size), with a guard column and Krudcatcher (Phenomenex, Torrance, CA, USA) maintained at 50 °C in a ThermoFisher Dionex UltiMate 3000 LC system (Thermo Fisher Scientific, UK). The injection volume was set to 10 μL, with samples held at 10 °C. LC mobile phases consisted of the following: A: 0.1% ammonium acetate in 60:40 water:acetonitrile; B: 0.1% ammonium acetate in 10:10:80 water:acetonitrile:isopropanol. Tandem MS/MS was performed using a Q-Exactive Plus Orbitrap MS (Thermo Fisher Scientific, UK), acquiring data simultaneously in full (*m*/*z* 200–2000; resolution 70,000) and MS/MS modes in both positive and negative ESI. Tandem MS/MS spectra were produced on the 5 most intense ions at any one time, at a resolution of 17,500. The flow rates of sheath gas, desolvation gas and sweep gas were 47.5, 11.25 and 2.25 units, respectively. The capillary and desolvation heater temperatures were set to 256 °C and 412 °C, respectively, and spray voltage set to 3500 V. Lipid identifications were made through database searches using LipidSearch software (Thermo Fisher Scientific, Horsham, UK).

### 2.7. Cholesterol Assay

Cholesterol levels within the cell-based lipid extracts were measured using the Amplex Red Cholesterol Assay Kit (Invitrogen, Horsham, UK; A12216). For each sample, 35 μL of 1× reaction buffer was pipetted into the 96-well plate to which 5 µL of sample was added. Then, 10 μL of 200 U/mL catalase (Sigma; C1345) was added to the samples and incubated at 37 °C for 15 min to remove background peroxides, based on the recommendations of Robinet et al., 2010 [28]. A mixture of the Amplex Red reagent, HRP, cholesterol oxidase, and/or cholesterol esterase in 1× reaction buffer was prepared according to the manufacturer’s instructions and used to measure total and free cholesterol levels. Reactions were incubated at 37 °C for 30 min whilst protected from light, after which fluorescence was measured at wavelengths of 544-nm excitation and 590-nm emission using a FLUOstar Omega microplate reader (BMG LABTECH).

### 2.8. Statistical Analyses

Two-sample statistical comparisons using *t*-tests and analysis of variance (ANOVA) were conducted within GraphPad Prism software. Gene expression and ontology analyses were implemented within R software using the *limma* [29] and topGO [30] packages, respectively. Differentially expressed genes (DEGs) were identified by applying Bayesian linear models and corrected for multiple comparisons using the Bonferonni–Holm method. Metabolite differences were statistically assessed by application of *t*-tests within MetaboAnalyst version 4.0 (Wishart Research Group, University of Alberta, Edmonton, Canada) [31].

## 3. Results

### 3.1. Paediatric Diffuse Glioma Cells Require Lipoproteins for Viability and Growth Maintenance

Cellular viability under lipoprotein-replete (normal medium; NM) or -deplete (lipoprotein-deplete medium; LPDM) culture conditions was assessed in adult U373 and paediatric KNS42 and SF188 models of diffuse gliomas over a 7-day period. Lipoprotein starvation did not significantly reduce the cellular viability of U373 cells (Figure 1A), in contrast to the greater detriment to growth observed in KNS42 and SF188 cells, as depicted by the loss of exponential growth in LPDM (Figure 1B,C). Significant differences in cellular viability were measured in KNS42 after 5 days of culture in LPDM and at the 3-day timepoint in SF188 cells.

Three lower-grade paediatric gliomas were also assessed to determine if the metabolic responses observed replicate those observed in the high-grade paediatric diffuse gliomacell lines. Growth of the grade III UW479 and grade II Res259 cell lines under lipoprotein-deplete conditions resulted in a higher PrestoBlue readout, compared to the control cells, which was significant at day 7 in both cases (*p* < 0.05) (Appendix A). Cell viability assessment using the crystal violet assay showed the opposite result, with reduced viability over time. In both cell lines, the difference in viability was significant at day 7 but was more prominent in the UW479 cell line, compared to the Res259 cell line (Appendix A). The grade I Res186 cell line, in contrast, demonstrated the largest response to growth under lipoprotein-deficient conditions, as determined by both the PrestoBlue and crystal violet assays (Appendix A). Control Res186 cells demonstrated a normal growth curve, whereas cells deprived of lipoproteins showed a minimal increase in growth. The effect of removing lipoproteins from the medium was likely cytostatic, considering that Res186 cells were still present within wells without any overt morphological indications of apoptosis induction (Appendix A). However, lipoprotein-starved Res186 cells did appear morphologically different compared to the control cells, characterised by an elongated phenotype. The findings indicate a higher capacity of paediatric glioma cell lines compared to their adult counterpart to increase reducing potential following metabolic stress, as indicated by the PrestoBlue assay. A possible explanation for these differences may be accounted for in the underlying tumour biology between age groups, as represented by these cell lines.

Differences between two- and three-dimensional cellular models regarding phenotypic manifestations are well documented [32]. The consequences of lipoprotein starvation were, therefore, studied in spheroid models of adult and paediatric diffuse glioma cell lines. Consistent with monolayer set-ups, U373 spheroids grew largely unhindered in LPDM (Figure 1D), in stark contrast to the KNS42 and SF188 cell lines, demonstrating significant differences in spheroid diameter between NM and LPDM culture conditions from the 9-day timepoint onwards (Figure 1E,F). KNS42 and SF188 spheroid diameters were minimally reduced from the point of lipoprotein starvation throughout a 14-day period of culture, indicative of a cytostatic response. Cell cycle analysis of monolayer-cultured adult GBM cells at the 3-day timepoint revealed an unaltered profile for U373 in LPDM (Figure 1G), whereas a significantly increased percentage of paediatric diffuse glioma cells in S phase and decreased percentage of G0/1 phase cells were observed (Figure 1H,I), with a concomitant reduction in cells in the G2/M phase for the KNS42 cell line only (Figure 1H). These findings support an elongation of the S phase in paediatric diffuse glioma cells grown in the absence of lipoproteins. Nevertheless, based on the known inter-tumour heterogeneity, which manifests in adult GBM, we cannot generalise that there are intrinsic differences in response to lipoprotein deficiency between paediatric and adult high-grade glioma, based upon the U373 adult cell line alone.

### 3.2. Cholesterol-Related Processes Define Diffuse Glioma Response to Lipoprotein Deprivation 

Phenotypic manifestations, resulting from lipoprotein deprivation, were particularly evident in paediatric diffuse glioma cell lines from the 3-day timepoint onwards, supported by the hierarchical clustering of the microarray transcriptomes of KNS42 and SF188 cell lines (Figure 2A). Determination of differentially expressed genes (DEGs) after lipoprotein starvation for 3 days was conducted to elucidate the transcriptomic response that preceded overt indications of reduced cellular viability and growth. Fifty-one significant DEGs were identified for the U373 GBM cell line after correction for multiple testing. Cellular response to hypoxia was identified as the top enriched process by GO enrichment analysis, despite cell culture in normoxia, given the upregulation of STC2, CA9, BNIP3, VEGFRA, and NDNF (Table 1). Upregulation of the extracellular enzyme LIPG, which facilitates lipoprotein uptake and lipid release from HDLs, was indicative of cellular response to lipoprotein deficiency [33]. Gene associations related to sterol metabolism in the U373 GBM cell line were revealed by network analysis (Network Analyst21 on DEGs, with an adjusted *p*-value of 0.05 (total of 547 genes) (Appendix A).

Of the 197 and 565 significant DEGs identified in KNS42 and SF188 cells, respectively, the induction of inflammatory responses was supported by the enrichment of the GO terms type I interferon signalling pathway and defence to virus (Table 1). Network analysis of DEGs in these paediatric diffuse glioma cells revealed a module of interconnected key inflammatory genes, including STAT1, STAT2, and IRF9 (Appendix A). As observed for the U373 GBM cell line, cholesterol dysregulation was supported by the upregulation of several cholesterol-related transcripts, including HMGCS1, HMGCR, MVK, DHCR24 and DHCR (Figure 2B and Appendix A), consistent with the enrichment of the GO term cholesterol biosynthetic process (Table 1). Gene expression alterations in several genes associated with cholesterol synthesis, regulation, export/import and storage indicated a greater fold-change increase in paediatric diffuse glioma cells, compared to the U373 GBM cell line (Figure 2B). Of note, the fold-change decrease in ABCA1 was much greater in magnitude in U373 cells compared to the paediatric diffuse glioma cell lines (Figure 2B), implicating reduced export as the primary mechanism of cholesterol homeostatic maintenance in U373 cells, as opposed to an increase in de novo synthesis. Finally, to establish putative prognostic indications, genome-wide gene expression and survival data for adult primary GBM available via The Cancer Genome Atlas was used to investigate the top DEGs induced in U373 cells STC2, CA9, BNIP3, VEGFRA, and NDNF upon lipoprotein deprivation. Only CA9 was associated with survival, whereby the CA9 high expression group had a significantly poor overall survival outcome (*p*-value = 0.049) compared to the CA9 low expression group (Figure 2C). 

### 3.3. Taurine and Choline Metabolism Are Perturbed upon Removal of Exogenous Lipoproteins

We next investigated the consequential effects of lipoprotein starvation on intermediary metabolism as the underlying basis of reduced cellular viability and growth. Liquid chromatography–mass spectrometry (LC–MS)-based metabolite profiling was conducted, leading to the elucidation of 604 putatively identified human metabolites. Of the glycolytic intermediates identified, only glyceraldehyde 3-phosphate and phosphoenolpyruvate displayed significant alterations (increased and decreased levels, respectively) in lipoprotein-deprived U373 GBM cells, with pyruvate as the sole metabolite to be significantly reduced in paediatric diffuse glioma cell lines, suggestive of reduced synthesis or increased shuttling into the tricarboxylic acid (TCA) cycle (Figure 3A; Appendix A). Dysregulated mitochondrial metabolism, as indicated by reduced succinate levels in U373 and KNS42 but not SF188 cells, was determined by TCA cycle metabolite analysis (Figure 3B; Appendix A). Significantly reduced malate levels were also observed for both SF188 and KNS42 cells upon lipoprotein starvation, suggesting insufficient replenishment of metabolite pools (Figure 3B; Appendix A). Adenosine 5′-triphosphate (ATP), the energetic product of glycolysis and oxidative phosphorylation, was significantly increased in lipoprotein-starved U373 GBM cells, whereas no significant changes were identified in the paediatric diffuse glioma cell lines (Appendix A).

The culture of adult and paediatric diffuse gliona cells in LPDM caused only low positive fold-changes in most amino acids identified, inconsistent with the marked metabolic response and only reaching significance consistently in the SF188 cell line (Figure 3B; Appendix A). Given the role of glutamine in TCA cycle anapleurosis, it was surprising that significantly increased, rather than decreased, levels of glutamine were observed in KNS42 and SF188 cells (Figure 3B; Appendix A). Within the methionine cycle and downstream transsulphuration pathway, significantly reduced levels of serine and cystathionine were observed in all three cell lines, in addition to a significant increase in glycine and methionine levels in SF188 cells (Figure 3C; Appendix A). The end-product of the pathway glutathione and its oxidised form were not significantly altered in any of the three cell lines (Figure 3C; Appendix A), indicating lack of stress induced by reactive oxygen species (ROS). The online MetaboAnalyst metabolomics analysis tool was used to detect enriched metabolite sets and pathway impact. The only metabolite set to reach significance at the raw *p*-value level in SF188 and KNS42 cells was taurine and hypotaurine metabolism (0.0069 and 0.02, respectively), whereas no metabolite set reached the significance threshold for the U373 cell line. Significant reduction in cysteine sulfinate, hypotaurine and taurine (metabolites of the taurine biosynthesis pathway) indicated the detrimental impact of lipoprotein deprivation on taurine metabolism (Figure 3C and Appendix A; Appendix A).

Increased levels of proline reached significance in U373 and SF188 cells, whereas its immediate catabolic breakdown product 1-pyrroline 5-carboxylate (P5C) increased significantly in KNS42 and SF188 cells (Figure 3D; Appendix A). Consistent with a possible membrane lipid synthetic response following lipoprotein deprivation, choline levels were consistently reduced in all cell lines to a significant level, but choline phosphate levels were only significantly reduced in U373 cells (Figure 3E; Appendix A). Interestingly, levels of cytidine diphosphate (CDP)-choline, used in the synthesis of phosphatidylcholine (PC), were significantly increased in lipoprotein-deprived KNS42 and SF188 cells (Figure 3E; Appendix A). The most prominent change to nucleotide metabolism was the association of high levels of xanthine with low levels of uric acid in all diffuse glioma cell lines, with *p*-values reaching the significance threshold (Figure 3F; Appendix A), indicating reduced conversion of xanthine into uric acid under lipoprotein-deplete conditions.

### 3.4. Lipoprotein Starvation Induced Disturbances to Lipid and Cholesterol Species

The role of lipoproteins as a source of lipids and cholesterol species led us to conduct MS-based lipidomics of lipoprotein-starved GBM cells, leading to the elucidation of the increased abundance of several PC and phosphatidylethanolamine species in all three cell lines (Figure 4A–C). Of note, an almost equal number of increased and decreased PC species was present in the KNS42 cell line (Figure 4B). Several triglyceride species were reduced following lipoprotein starvation in all cell lines, but this was more pronounced in the paediatric diffuse glioma cell lines, especially SF188 (Figure 4C). The implied alterations to lipid droplet dynamics, following the observation of several reduced triglyceride species and considering the cholesterol component of lipoproteins, led us to assess the levels of cholesterol in free or esterified form. All three diffuse glioma cell lines demonstrated a significant drop in total cholesterol levels (Figure 4D), but significantly reduced levels of free cholesterol were only observed in KNS42 cells (Figure 4E). The change in cholesterol ester levels was negative in U373, KNS42, and SF188 cells, indicating a reduction in cholesterol ester content under lipoprotein-deplete conditions, reaching significance in all three diffuse glioma cell lines (Figure 4F).

### 3.5. LXR Agonists

A reduction in cholesterol levels was observed in adult and paediatric diffuse glioma cells cultured under lipoprotein-deplete conditions, possibly underlying the transcriptional induction of genes related to cholesterol regulation and synthesis. These findings suggest that adult and paediatric diffuse glioma cells harbour a metabolic vulnerability if cholesterol regulation and metabolism are targeted. We, therefore, pharmacologically replicated lipoprotein deprivation and stimulated cholesterol export to assess the therapeutic efficacy and mode of action of LXR-623, a partial LXRα/full LXRβ agonist, in these in vitro models. Treatment of U373, KNS42, and SF188 cells with LXR-623 over 72 h indicated that the detrimental effects of LXR agonist treatment manifest following at least 24 h exposure (Figure 5A–C). Dose-response curves of LXR-623 indicated half-maximal inhibitory concentrations (IC50) of 8.50 μM, 27.51 μM, and 22.49 μM in U373, KNS42 and SF188 cells, respectively (Figure 5D–F). Cell cycle analyses was conducted on diffuse glioma cells treated both at IC50 or IC75 LXR-623 concentrations based on the dose-response curves. A significant increase and decrease in the percentage of S and G2/M phase cells, respectively, were shown by U373 exposure to LXR-623 at IC50 and IC70 concentrations (Figure 5G). In contrast, a significant increase and decrease in the percentage of sub G0 and G0/G1 phase cells, respectively, was observed in KNS42 and SF188 cells exposed to IC50 and IC70 concentrations (Figure 5H,I).

Interestingly, addition of exogenous cholesterol (at 5 µM and 10 µM concentrations) significantly rescued LXR-623-treated U373 GBM cells (Figure 6A), highlighting reduced intracellular cholesterol as plausibly contributing to the underlying defect, but did not rescue the cellular viability of KNS42 or SF188 cells (Figure 6B,C). It is notable that supplementation with 20 μM MβCD-cholesterol caused a significant reduction in cellular viability in all three diffuse glioma cells lines. Whilst this may indicate toxic accumulation of free cholesterol, we cannot exclude direct cell membrane damaging induced by MβCD-cholesterol at this high concentration.

GO processes associated with cellular signalling, proliferation and apoptosis were enriched in U373 cells exposed to the IC50 concentrations of LXR-623 for 72 h (Appendix A), as reflected in the network of differentially expressed genes in Figure 7A. Interestingly, increased expression of ATF alludes to ER stress induced upon LXR-623 treatment (Figure 7A). In contrast, GO processes associated with DNA replication, cell cycle and cell division were commonly enriched in both KNS42 and SF188 cells (Appendix A), as supported by large networks of downregulated genes with roles in cell cycle regulation (Figure 7B,C).

It was particularly evident from the transcriptomics data that upregulation of a cholesterol biosynthetic response was a key feature of paediatric diffuse glioma cells under lipoprotein-deficient conditions. Examination of the changes in expression of several genes involved in cholesterol regulation, synthesis, export, import and storage revealed different responses between the adult U373 and the paediatric KNS42 and SF188 cell lines. Paediatric diffuse glioma cell lines demonstrated a positive log2 fold change in most genes involved in regulation, synthesis, import and storage of cholesterol under lipoprotein-deficient conditions. For most genes, a positive log2 fold change was also identified for the U373 cell line, but was lower in magnitude compared to the paediatric diffuse glioma response. Interestingly, the negative log2 fold change in ABCA1 was much greater in magnitude in the U373 cell line compared to the paediatric diffuse glioma cell lines, suggesting that reduced export is the primary mechanism by which U373 cells maintain cholesterol homeostasis under lipoprotein-deficient conditions (Appendix A).

## 4. Discussion

The seminal observations made by Otto Warburg in the 1920s regarding aerobic glycolysis has fuelled scientific interest in the study of cancer metabolism [32]. Almost a century on, reprogramming of normal cellular metabolism to meet energetic and biosynthetic needs is now considered a hallmark of cancer [33]. In cancer, malignant progression is often dependent on the acquisition of a lipogenic phenotype, in which uptake and endogenous synthesis of fatty acids is increased [34]. However, cancer cells demonstrate different capacities to induce de novo fatty acid synthesis under lipid-poor conditions [35] and can instead scavenge fatty acids from the extracellular milieu, especially under hypoxic conditions [36,37]. Here, we demonstrate a requirement of paediatric diffuse glioma cells on exogenous lipoprotein components to sustain viability and growth, with a comparatively less pronounced impact on the adult U373 GBM cell line. This does not conclusively indicate a paediatric diffuse glioma-specific phenotype, as growth of the adult U87 GBM cell line has been previously shown to be reduced under lipoprotein-deficient conditions [18], but supports further study in additional models of adult and paediatric diffuse gliomas. Furthermore, adult GBM inter-tumour heterogeneity warrants caution when interpretating comparative data when only one cell line has been used. However, we note that of the top DEG induced upon lipoprotein deprivation in the U373 adult GBM cell line, high *CA9* expression confers a significantly worse overall survival in the lipoprotein-replete conditions of primary tissue, as determined by the TCGA analyses. 

Mitogenic properties of HDLs has been observed in fibroblast models, with uptake of HDLs associated with cell cycle progression [38]. The reduced growth rate of the paediatric diffuse glioma cells and the cytostatic response of spheroids indicated altered cell cycle dynamics, evidenced by an increased fraction of cells in S phase. Whilst the transcriptomic signature at the same timepoint did not reveal GO terms enriched for cell cycle regulation processes, we observed an upregulation of genes associated with cholesterol biosynthesis, thus supporting accumulating evidence of lipoproteins as a tumour-promoting source of cholesterol in diffuse glioma [7,8,9]. 

GBM cells that express the oncogenic epidermal growth factor receptor variant, EGFRvIII, demonstrate a non-oncogene addiction to lipoproteins due to increased LDLR-mediated uptake [9]. We identified an upregulation in the expression of *LDLR* and *VLDLR* in the transcriptomic response to lipoprotein deprivation, likely regulated via the observed induction of the sterol regulatory axis initiated by *INSIG1/2*. Lipoprotein starvation in the paediatric diffuse glioma cells surprisingly induced a set of genes associated with type I interferon (IFN) signalling, normally associated with a viral response. A decreased pool size of synthesised cholesterol has been documented to engage a type I IFN response in macrophages [39]. This metabolic-inflammatory circuit may exist in KNS42 and SF188 cells, as reflected in our GO analysis, possibly alluding to an inherent attribute of astrocytes, given their ability to respond to viral infection of the CNS through IFNAR signalling [40]. Additionally, we observed an increase in the expression of *IRF9* and *STAT2* of the IFN signalling pathway within the paediatric diffuse glioma cell lines, suggesting activation of an inflammatory stress response shown to decrease chemosensitivity in small cell lung cancer cells [41]. IRF9 and unphosphorylated STAT2 cooperate with NF-kB to induce IL-6 expression [42], a cytokine that promotes tumour growth of EGFR-driven GBM in an autocrine and paracrine manner [43]. Increased expression of IFN pathway genes may, therefore, constitute a stress response within the two pGBM cell lines that, under the right context (e.g., chemotherapeutic insult), may promote tumour survival for short periods of stress.

The culture of cancer cells under nutrient-deplete conditions, most commonly glucose, highlights the ability of cancer to rely on different pathways of intermediary metabolism to meet cellular demands for energetic substrates, anabolic building blocks, and redox species [44]. Here, we utilised LC–MS to demonstrate evidence of aberrant mitochondrial metabolism and the significant perturbation of taurine metabolism in the paediatric diffuse glioma cell lines starved of lipoproteins. Cholesterol depletion in tumour cells has been documented to reduce taurine uptake and cellular content, leading to reduced cell volumes, given the role of taurine as an organic osmolyte [45]. Interestingly, lower relative levels of uric acid and the accumulation of its immediate precursor, xanthine, in all three diffuse glioma cell lines hinted at reduced xanthine oxidase activity in lipoprotein-deplete conditions, possibly to reduce ROS generation and resultant oxidative damage. The oncogenic and/or tumour suppressive functions of xanthine oxidase remain contested [46], but a role for xanthine oxidase as a downstream effector within a metabolic-inflammatory circuit, initiated by the nuclear transcriptional coactivator function of hexokinase 2, has been observed in glucose-starved glioma cells [47].

The identification of reduced triglyceride and total cholesterol levels in all three lipoprotein-starved diffuse glioma cell lines led us to investigate LXR agonists, which were initially developed to reduce cardiovascular disease by preventing lipoprotein uptake via induction of *MYLIP* (which encodes IDOL), an E3 ubiquitin ligase that mediates the degradation of LDLR and VLDLR, and by promoting cholesterol export via ABCA1 and ABCG1 membrane transporters [48]. The therapeutic potential of LXR agonists has been demonstrated using EGFRvIII-expressing GBM models in vitro and in vivo [8,9]. Here, we show their first application in paediatric diffuse glioma models that are not characterised by EGFRvIII, although convergent activation of downstream signalling effectors within the PI3K/AKT–SREBP–LDLR axis may confer a similar non-oncogene addiction to lipoproteins. The addition of exogenous cholesterol was sufficient to rescue viability in U373 GBM cells but not in paediatric diffuse glioma cells, indicating that cholesterol-independent mechanisms may have more import in mediating LXR agonist toxicity in KNS42 and SF188 cells, as opposed to cholesterol export. Increased expression of *ATF4* implicated a mechanism of cell death associated with ER stress, a process induced upon altered lipid dynamics [49], which may represent a mechanism to negate cholesterol export to HDL [50]. Pharmacological means of decreasing cellular cholesterol through statin use, for example, have proven to be effective in reducing unrestrained oncogenic growth [51], and have been associated with reduced glioma risk in a large epidemiological study of 2656 cases and 18,480 controls [52]. Therapeutic levels of LXR-623 are reached within the CNS in mouse models and LXR-623 activation in preclinical GBM models has recently been shown not to suppress oxidative phosphorylation in healthy astrocytes [53]; however, the identification of nervous system disorders in a phase I/II clinical trial precludes further clinical investigation of this compound in GBM. Nevertheless, our findings represent proof-of-concept for the consideration of combination strategies, enabling lower concentrations of LXR agonists to be used in conjunction with standard treatment regimens, which may be more therapeutically feasible [54,55], particularly in the paediatric setting in which the CNS is still developing.

## 5. Conclusions

Whilst the therapeutic potential of LXR agonists has been demonstrated using EGFRvIII-expressing adult glioblastoma models in vitro and in vivo, here, we show a first application in paediatric diffuse glioma models that are not characterised by EGFRvIII mutation. Pharmacological reduction in intracellular cholesterol via decreased uptake and increased export was simulated using the liver X receptor agonist LXR-623, which reduced cellular viability in paediatric diffuse glioma models.

Our findings represent proof-of-concept for the consideration of combination strategies, enabling lower concentrations of LXR agonists to be used in conjunction with standard treatment regimens, which may be more therapeutically feasible, particularly in the paediatric setting in which the CNS is still developing.

## Figures and Tables

**Figure 1 cancers-14-03873-f001:**
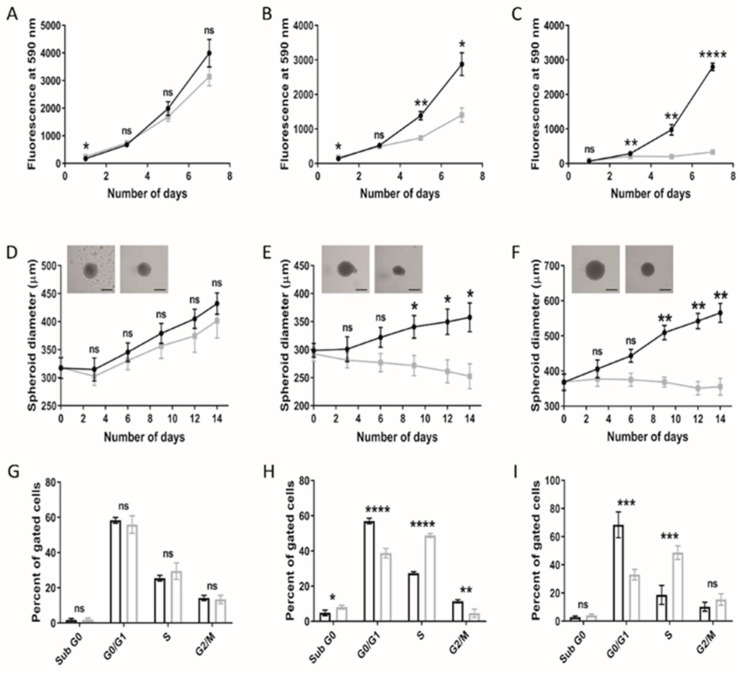
**Altered growth characteristics of adult and paediatric diffuse glioma cells under lipoprotein-deplete conditions.** U373 (adult GBM), KNS42 and SF188 (both paediatric diffuse glioma) cells were grown under lipoprotein -replete (black) or -deplete (grey) conditions in monolayer (**A**–**C**) or spheroid (**D**–**F**) format, respectively, over a 7- or 14-day period, respectively. (**D**–**F**) Brightfield images are representative of 14-day old spheroids in lipoprotein-replete (left) or -deplete (right) conditions. Black bars depict 300 μm in scaled length. (**G**–**I**) Flow cytometric cell cycle analysis following U373, KNS42 and SF188 monolayer culture, respectively, under lipoprotein -replete (black) or -deplete (grey) conditions for 72 h. Results are the mean ± SEM (**A**–**F**) or SD (**G**–**I**) for n ≥ 3 independent replicates (n = 6 spheroid replicates). Statistical evaluation of differences between lipoprotein -replete or -deplete conditions was conducted using *t*-tests: ns = not significant * *p* < 0.05; ** *p* < 0.01; *** *p* < 0.001; **** *p* < 0.00001.

**Figure 2 cancers-14-03873-f002:**
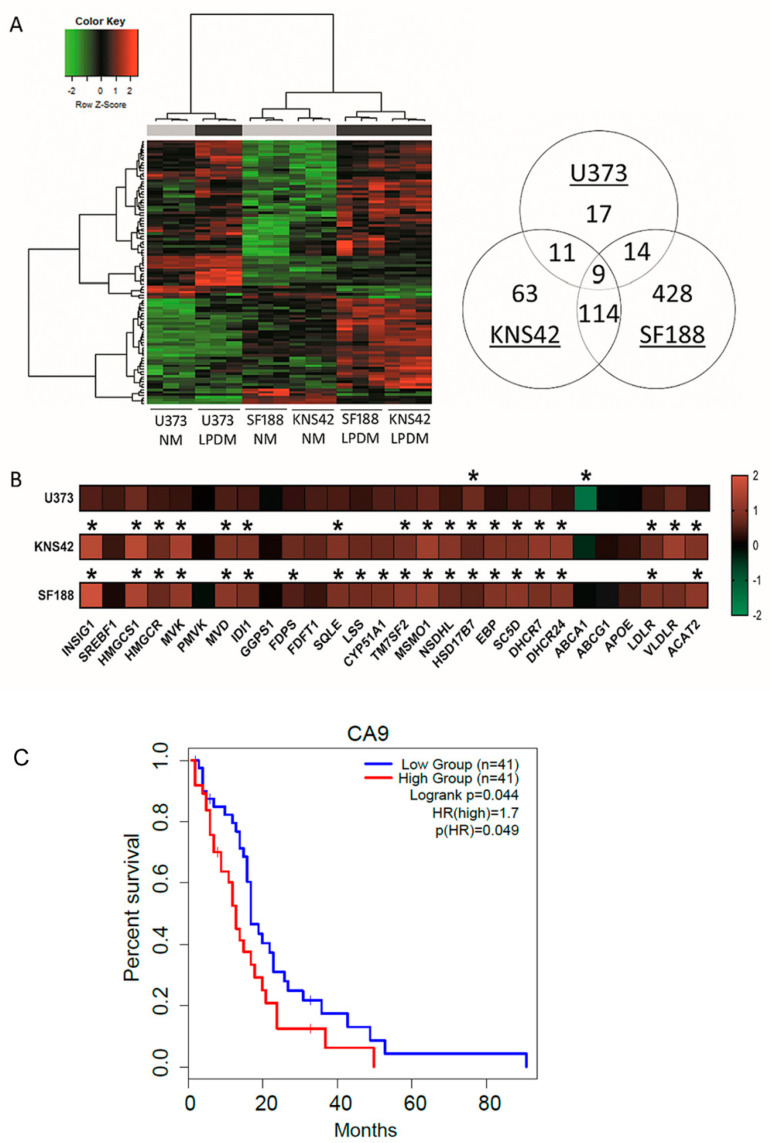
**Cholesterol metabolism-related transcriptomic signatures induced by lipoprotein starvation.** (**A**) Hierarchical clustering analysis of U373, KNS42, and SF188 diffuse glioma cells cultured under lipoprotein-replete (grey bar) or -deplete (black bar) conditions. Venn diagram (right) indicates the number of differentially expressed genes under lipoprotein-deplete conditions shared across the three diffuse glioma cell lines. (**B**) Heatmap representation of cholesterol metabolism-related gene expression for each of the three diffuse glioma cell lines cultured under lipoprotein-deplete conditions. Colour scale represents log2 fold change gene intensity values. Results are the mean of n = 3 independent replicates. Statistical evaluation of differences between lipoprotein -replete or -deplete between the two culture conditions was performed by implementing a Bayesian linear model using the R package *limma* and correcting for multiple comparisons: * *p* < 0.05. (**C**) Kaplan–Meier curve representing the overall survival outcome between *CA9*-High and *CA9*-Low groups based upon TCGA adult GBM data.

**Figure 3 cancers-14-03873-f003:**
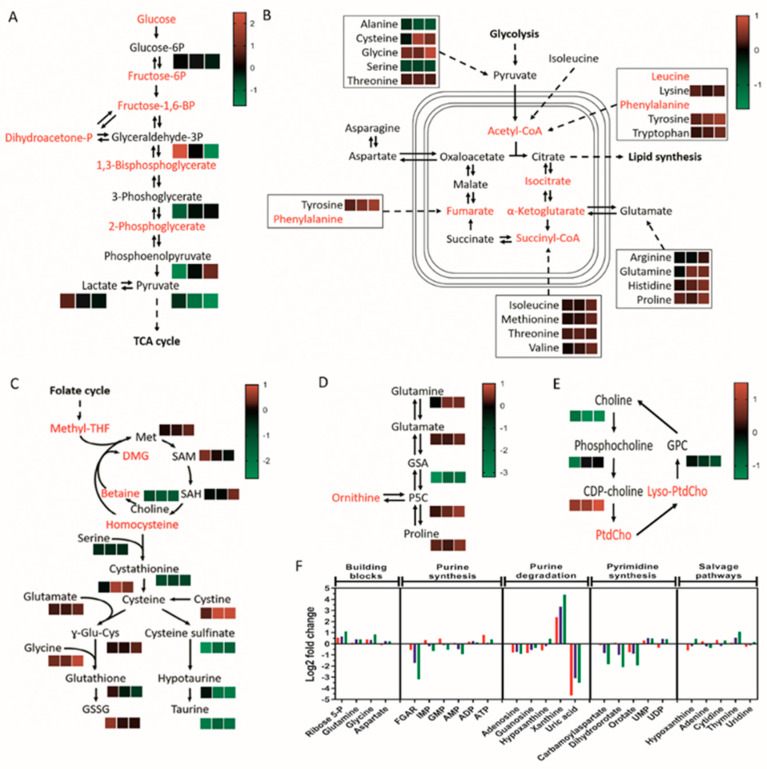
**Lipoprotein starvation-induced perturbation of central metabolic pathways.** Identified intermediates within glycolysis (**A**), TCA cycle and linked amino acid pathways (**B**), methionine cycle and transulphuration pathway (**C**), proline metabolism (**D**), choline pathway (**E**), and nucleotide metabolism (**F**) are depicted in black, with unidentified intermediates in red. (**A**–**E**) Heatmaps represent peak intensity fold changes between lipoprotein-replete and -deplete conditions in U373 (left box), KNS42 (middle box), and SF188 (right box) cells. Colour scales indicating the fold change range are specific to each panel. (**F**) Log2-fold peak intensity changes in nucleotide metabolism intermediates in U373 (red), KNS42 (blue), and SF188 (green) cells. Results are the mean of n = 6 independent replicates.

**Figure 4 cancers-14-03873-f004:**
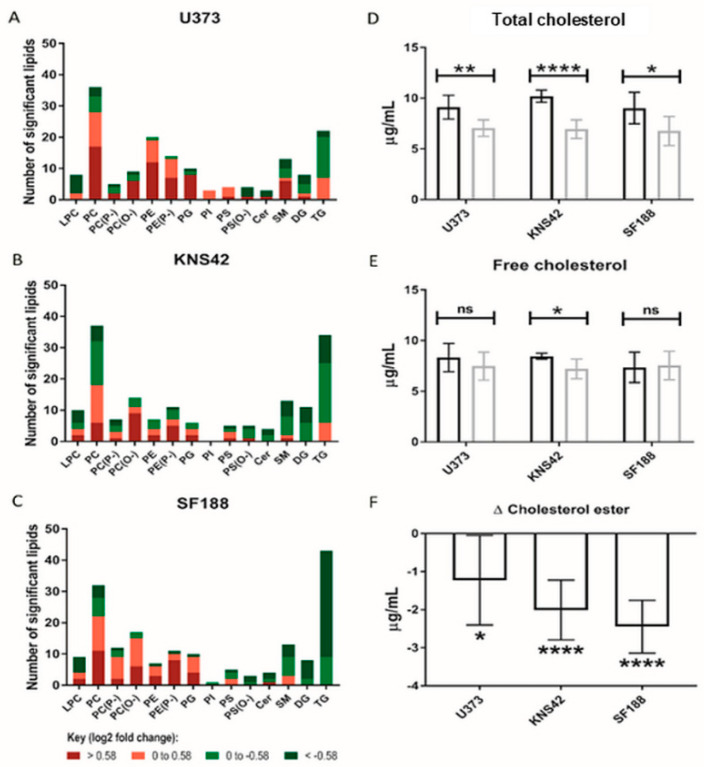
**Lipid and cholesterol alterations under lipoprotein-deplete conditions.** (**A**–**C**) Number of each lipid species significantly altered between lipoprotein-replete and -deplete conditions in U373, KNS42, and SF188 diffuse glioma cells, respectively. Colours represent different categories of log2-fold peak intensity changes, with dark red and dark green representing fold changes of 1.5 and 0.67, respectively. (**D**,**E**) Measurement of the total and free cholesterol concentrations in diffuse glioma cells under lipoprotein-replete (black) and -deplete (grey) conditions. (**F**) The change in concentration of cholesterol ester in each cell line upon lipoprotein starvation was calculated from the concentrations of total and free cholesterol. Results are the mean ± SD of n = 6 independent replicates. Statistical evaluation of differences between the two culture conditions was conducted using *t*-tests: ns = not significant * *p* < 0.05; ** *p* < 0.01; **** *p* < 0.00001. Abbreviations: LPC: lysophosphatidylcholine; PC: phosphatidylcholine; PE: phosphatidylethanolamine; PG: phosphatidylglycerol; PI: phosphatidyl-inositol; PS: phosphatidylserine; Cer: ceramide; SM: sphingomyelin; DG: diacylglycerol; TG: triacyl-glycerol. (P−) and (O−) represent alkenyl-acylphospholipids (plasmologens) and alkyl-acylphospho-lipids, respectively.

**Figure 5 cancers-14-03873-f005:**
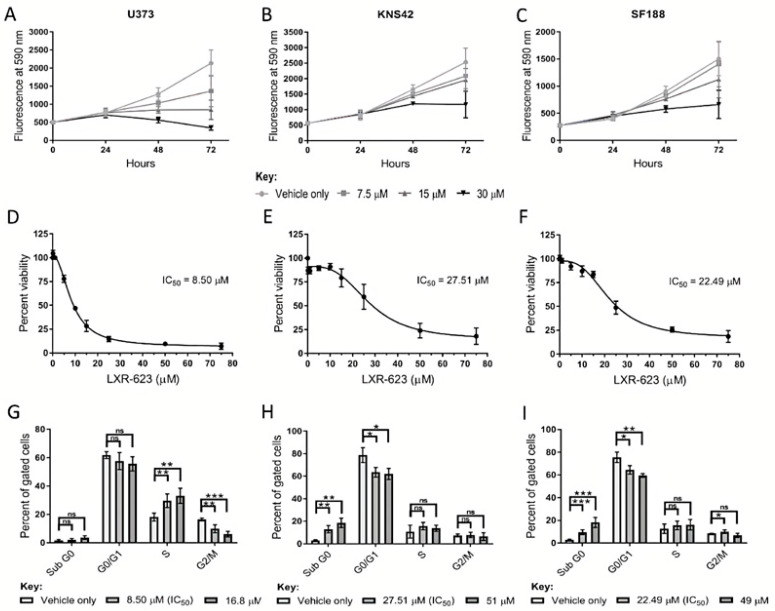
**Therapeutic efficacy of the LXR agonist LXR-623 in diffuse cells.** (**A**–**C**) Treatment of U373, KNS42, and SF188 cells with 7.5, 15, and 30 µM LXR-623 for 72 h, respectively. (**D**–**F**) Dose-response curves to determine IC_50_ (half maximal inhibitory concentration) values of LXR-623 in U373, KNS42, and SF188 cells. (**G**–**I**) Flow cytometric cell cycle analysis following treatment with LXR-623 for 72 h at respective IC_50_ concentrations and at higher concentrations deemed to result in 25% cellular viability. Results are the mean ± SEM (**A**–**F**) or SD (**G**–**I**) for n ≥ 3 independent replicates. Statistical evaluation of differences between the two culture conditions was conducted using *t*-tests: ns = not significant * *p* < 0.05; ** *p* < 0.01; *** *p* < 0.001.

**Figure 6 cancers-14-03873-f006:**
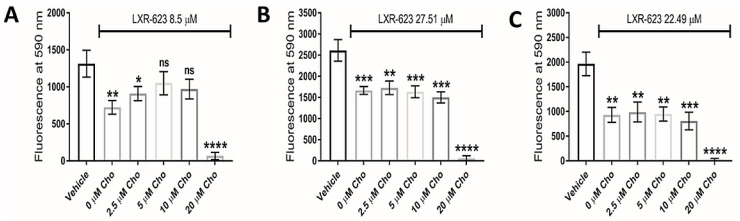
**Dependence of LXR-623 on cholesterol export for therapeutic efficacy.** (**A**–**C**) Cellular viability of U373, KNS42 and SF188 glioma cells, respectively, treated with LXR-623 for 72 h in the presence of different concentrations of water-soluble cholesterol (MβCD-Cho). Results are the mean ± SEM for n = 3 independent replicates. Statistical evaluation of was conducted using ANOVA: ns = not significant * *p* < 0.05; ** *p* < 0.01; *** *p* < 0.001; **** *p* < 0.00001.

**Figure 7 cancers-14-03873-f007:**
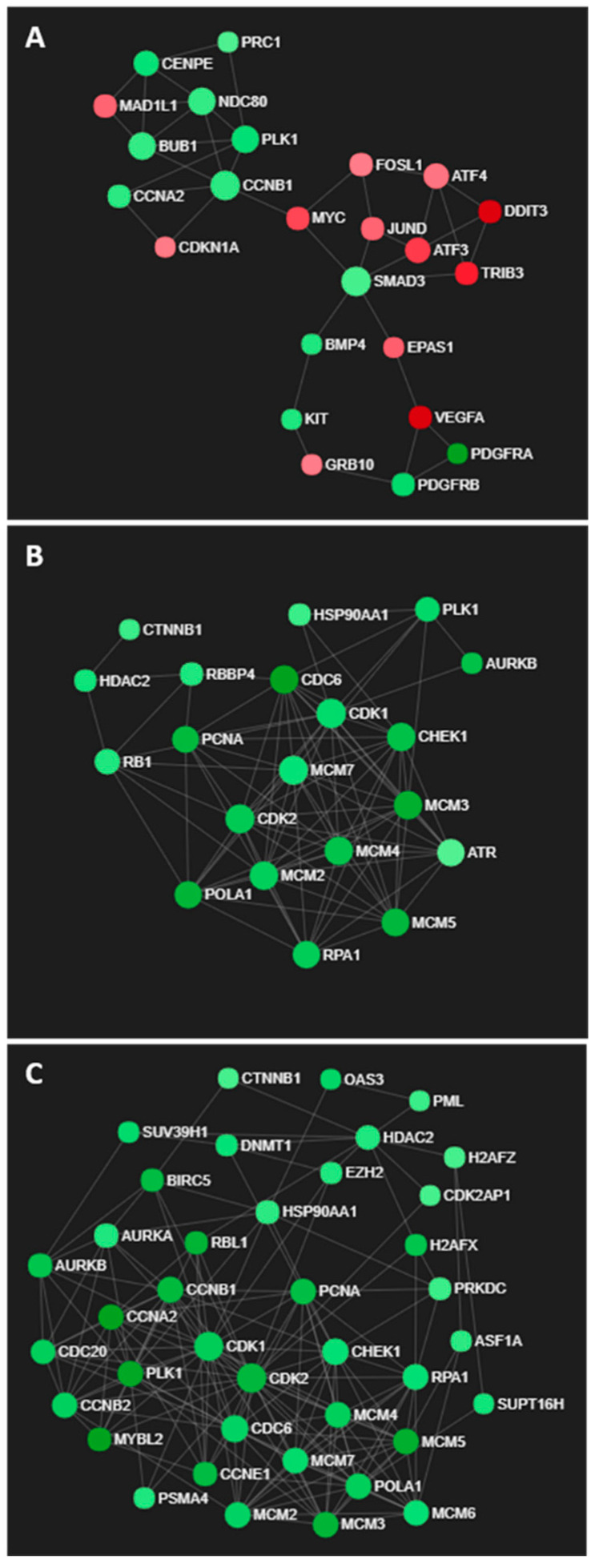
Transcriptomic networks induced by LXR-623 treatment. (**A**) Interconnected network of ER stress- and cell cycle-related genes in LXR-623-treated U373 cells. (**B**,**C**) Networks of downregulated genes associated with cell cycle regulation in LXR-623-treated KNS42 and SF188 cells, respectively.

**Table 1 cancers-14-03873-t001:** Gene ontology analysis of differentially expressed genes in U373, KNS42 and SF188 diffuse glioma cells starved of lipoproteins.

GO.ID	Term	Annotated	Significant	Expected	classicFisher	Weight01KS	Genes
**U373**							
GO:0071456	Cellular response to hypoxia	98	5	0.53	1.70 × 10^4^	8.80 × 10^4^	*STC*, *CA9*, *BNIP3*, *VEGFA*, *NDNF*
GO:0032376	Positive regulation of cholesterol transport	11	2	0.06	1.52 × 10^3^	2.69 × 10^2^	*ABCA1*, *LIPG*
GO:0030823	Regulation of cGMP metabolic process	12	2	0.06	1.82 × 10^3^	3.50 × 10^2^	*VEGFA*, *PDE5A*
GO:0030199	Collagen fibril organisation	15	3	0.08	6.40 × 10^5^	4.81 × 10^2^	*LOX*, *LUM*, *P4HA1*
**KNS42**							
GO:0060337	Type 1 interferon signalling pathway	52	16	0.99	8.00 × 10^−16^	2.10 × 10^9^	*IFI6. OAS2*, *IRF7*, *OAS1*, *MX1*, *XAF1*, *IFI35*, *IFITM1*, *IRF9*, *IFI27*, *USP18*, *OAS3*, *ISG15*, *STAT2*, *IFIT3*, *IFIT2*
GO:0051607	Defence response to virus	107	17	2.05	1.30 × 10^−11^	2.00 × 10^5^	*DDIT4*, *OAS2*, *IRF7*, *OAS1*, *IFI44L*, *MX1*, *IFITM1*, *IRF9*, *OAS3*, *ISG15*, *IFNE*, *CXCL10*, *STAT2*, *HTRA1*, *IFIT3*, *IFIT2*, *MICA*
GO:0006695	Cholesterol biosynthetic process	42	15	0.8	5.10 × 10^−16^	8.00 × 10^3^	*INSIG1*, *HMGCS1*, *MSM01*, *DHCR24*, *MVK*, *DHCR7*, *NSDHL*, *ACAT2*, *SQLE*, *MVD*, *SC5D*, *EBP*, *IDI1*, *TM7SF2*, *HMGCR*
**SF188**							
GO:0060337	Type 1 interferon signalling pathway	55	20	3.08	4.90× 10^−12^	1.30 × 10^9^	*IFI6*, *IFIT1*, *OAS2*, *XAF1*, *STAT2*, *MX1*, *IFIT2*, *IFIT3*, *IRF7*, *IRF9*, *MIR21*, *USP18*, *OAS3*, *ISG15*, *WNT5A*, *IFITM1*, *STAT1*, *SAMHD1*, *OASL*, *IFI35*
GO:0045540	Regulation of cholesterol biosynthetic process	27	14	1.51	2.50 × 10^−11^	1.10 × 10^7^	*HMGCS1*, *MVK*, *SC5D*, *TM7SF2*, *DHCR7*, *MVD*, *SQLE*, *CYP51A1*, *IDI1*, *LSS*, *SCD*, *HMGCR*, *FDPS*, *FASN*
GO:0051607	Defence response to virus	111	28	6.22	7.30 × 10^−12^	1.80 × 10^6^	*IFIH1*, *IFIT1*, *OAS2*, *IL1B*, *STAT2*, *MX1*, *IFIT2*, *IFIT3*, *IL6*, *IRF7*, *IFI44L*, *PML*, *IRF9*, *HERC5*, *DDX58*, *OAS3*, *MICA*, *ISG15*, *GBP3*, *CXCL10*, *IFITM1*, *TNFAIP3*, *STAT1*, *HTRA1*, *SAMHD1*, *PARP9*, *OASL*, *EXOSC5*
GO:0006695	Cholesterol biosynthetic process	41	20	2.3	5.40× 10^−15^	2.80 × 10^3^	*INSIG1*, *HMGCS1*, *MSMO1*, *MVK*, *ACAT2*, *SC5D*, *TM7SF2*, *EBP*, *DHCR7*, *MVD*, *SQLE*, *CYP51A1*, *DHCR24*, *IDI1*, *LSS*, *SCD*, *HMGCR*, *NSDHL*, *FDPS*, *FASN*
GO:0016126	Sterol biosynthetic process	43	21	2.41	1.10 × 10^−15^	3.10 × 10^2^	*INSIG1*, *HMGCS1*, *MSMO1*, *MVK*, *ACAT2*, *SC5D*, *TM7SF2*, *EBP*, *DHCR7*, *MVD*, *SQLE*, *CYP51A1*, *DHCR24*, *IDI1*, *LSS*, *CYB5R2*, *SCD*, *HMGCR*, *NSDHL*, *FDPS*, *FASN*

Statistical evaluation of gene ontology (GO) categories based of the number of significant-to-expected observations using either Fisher’s exact test (Fis) or Kolmogorov–Smirnov test (KS). Classic and weight01 algorithms were utilised which either do or do not, take into account the structure of the GO hierarchy, respectively. Genes in bold were downregulated under lipoprotein-deplete conditions.

## Data Availability

Data is available (annotated IDEOM metabolomics data files and CEL gene expression microarray files for lipoprotein-replete and -deplete conditions) for sharing via access to the University of Nottingham Research Repository (https://rdmc.nottingham.ac.uk/), using reference: https://doi:10.17639/nott.7191.

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
