# Peer review of "Lipoprotein Deprivation Reveals a Cholesterol-Dependent Therapeutic Vulnerability in Diffuse Glioma Metabolism"

_cancers, 2022, doi:10.3390/cancers14163873_

Round 1

Reviewer 1 Report

The manuscript from Wood et al. represents an interesting study on the effects of cholesterol deprivation in glioma cell lines. While the study in principle is interesting and provides new insights into the role of lipid metabolism and the potential influenza of cholesterol deprivation on glioma cell viability and growth, some of the conclusion are rather farfetched and lack substantial evidence.

Please consider the following major concerns:

1)                  The authors only use three different cell lines – one cell line from an adult individual and two pediatric diffuse glioma cell lines. In these three cells line they observe the described differences in cholesterol contend and deprivation. This number of cell line it too small and too selective to make any generalized conclusion about the role of cholesterol deprivation in “adult” versus “pediatric” glioma in general. The authors may simply look at cell line specific differences. At least, the authors need to critically discuss the issue. The manuscript would be greatly improved if they also could include a lipidomics analysis from adult versus pediatric glioma tissues to show that this may resemble the differences in the lipid profile they find in the cell lines.

2)                  The authors observe, that the addition of exogenous cholesterol (at 5 µM and 10 µM concentrations) rescues LXR-623-treated U373 GBM cells. However, the differences shown in figure 6A are rather small. Is there really a specific underlying mechanisms or are they rather looking at cellular variability? Especially, since only the small number of n=3 experimenta has been performed. In addition, at 20 µM of MβCD-cholesterol they see an obviously toxic effect. There it is claimed this indicated a “toxic accumulation of free cholesterol”. While this may be true, no evidence is provided to support this conclusion. MβCD may rather exert a direct toxic, e.g. membrane damaging effect at this high concentration. This needs to be revised.

3)                  It is not exactly clear, how the different cell culture conditions were. In the Method section it is written “Cell lines were grown in Dulbecco’s Modified Eagle Medium (DMEM) supplemented with 10% foetal bovine serum (FBS)/lipoprotein deficient bovine calf serum LPDS; Bioquote; BT-907)…”. Throughout the manuscript, the authors only write “lipoprotein-replete/deplete conditions”. While they do not specify these conditions. Does this mean, the cells were cultered with 10% regular FBS versus another lipoprotein deficient bovine calf serum (=LPDS). If this is indeed the case, this would mean that the cell culture conditions would be very different between the “lipoprotein-replete/deplete” conditions, rendering the cell culture data rather inconclusive. This issue needs to be clarified.

Reviewer 2 Report

In this work, the authors are aimed at evaluating the potential impact of lipoprotein-derived cholesterol on glioma. The most relevant results demonstrated that lipoprotein depletion impacts glioma viability, and severely affects crucial metabolic processes, such as taurine metabolism. Similarly to lipoprotein depletion, intracellular cholesterol reduction by LXR agonists significantly reduces cell proliferation.

This work focuses on a hot topic in cancer biology, the experimental design is well conducted and the overall manuscript is well written.

However, there are some shortcomings that should be addressed:

1) The authors only use U373 cell line as experimental model of "adult glioma". It is not possible to draw any conclusion regarding prospective differences between "adult" and "paediatric" gliomas, since cell type-specific effects cannot be excluded. If the authors would compare adult and paediatric gliomas, they should include more than a single cell line. Otherwise, this consideration should be strongly scaled down in the whole manuscript.

2) In order to strengthen the validity of this work, some targets of interest derived from GO analysis (Table 1) should be validated by q-rtPCR expression.

3) Figure 4D: please correct the title of the graph

4) In some cases, data are expressed as the mean ± S.E.M. The standard error of the mean indicates the uncertainty of how the sample mean represents the population mean. In my opinion, the authors inappropriately report the SEM instead of the Standard Deviation (SD). Since the SEM is always less than the SD, it deceives the reader into underestimating the variability between independent experiments within the study sample.

Reviewer 3 Report

In this study, Wood et al. reported that LXR-623 could reduce the level of cholesterol of high-grade glioma cell, and thus impair cell growth. The validation of LXR-623-induced cytotoxicity implies a promising treatment strategy by targeting the metabolic requirement of diffuse glioma cell. Additional comments are listed here:

1. In vitro studies showed that LXR-623 could induce a significant toxicity to diffuse glioma cells, while whether LXR-623 is toxic to normal cell remain unclear.   

2. What is the mechanism of LXR-623 involved in reducing cholesterol level? Does LXR-623 reduce the cholesterol uptake or influence cholesterol metabolism?

3. Did the authors conducted any in vivo studies to evaluate the anti-glioma effect of LXR-623?

Round 2

Reviewer 1 Report

All my comments have been addressed sufficiently.

Reviewer 2 Report

Even though the authors did not perform some additional experiments required by the reviewer, they properly provided exhaustive explanations or alternatives to strengthen their data.

Despite this, I still have some concerns about the representation of the data which are often reported with SEM, while occasionally SD is also used in the manuscript. Notably SEM, an inferential parameter, quantifies uncertainty in the estimate of the mean. Differently, SD is a descriptive parameter and quantifies the variability. Since readers in biomedical field are generally interested in knowing the variability within the sample, the data shown in this manuscript should be precisely summarized with SD. Conversely, the use of SEM should be limited to compute the confidence interval, which measures the precision of population estimate. Again, the authors should substitute SEM with SD.